# Utility of cochlear perilymph biomarkers for hearing loss - A systematic review

**Kujani Wanniarachchi**[1☉], **Sita Tarini Clark** [2,3,4☉]*, **Neil Donnelly**[4], **Manohar Bance**[2,3,4], **Nathan Creber**[3,4,5,6]

1 School of Clinical Medicine, University of Cambridge, Cambridge, United Kingdom, 2 Department of Clinical Neurosciences, University of Cambridge, Cambridge, United Kingdom, 3 Cambridge Hearing Group, Cambridge, United Kingdom, 4 Department of Ear, Nose, and Throat Surgery, Addenbrooke's Hospital, Cambridge University Hospitals NHS Foundation Trust, Cambridge, United Kingdom, 5 Otolaryngology, Department of Surgery, The Royal Prince Alfred Hospital, Sydney, Australia, 6 University of Sydney, Australia

☉ These authors contributed equally to this work.
* stc47@cam.ac.uk

## Abstract

The live cochlea is a critical and delicate sensory organ with tissue sampling posing a risk to balance and hearing functions. Consequently, a researchers' ability to investigate the pathogenesis of sensorineural hearing loss (SNHL) has been limited, commonly involving in vitro models, translational animal models and post-mortem analysis. Previous human studies investigating SNHL have primarily been restricted to using peripheral blood or cerebrospinal fluid samples that indirectly measure inner ear pathologies. More recently, the establishment of a novel, safe, and feasible technique for intraoperative perilymph sampling has enabled the collection of a 'liquid biopsy' of the cochlea whilst still preserving inner ear function. This crucial development has opened avenues for characterising the cochlea microenvironment and subsequently investigating the mechanisms of SNHL. Perilymph sampling may also provide insights into why postoperative outcomes vary between cochlear implant users with otherwise similar preoperative clinical characteristics. The present systematic review aimed to elucidate the gap in the literature regarding the utility of perilymph biomarkers in SNHL diagnosis, treatment, and prognosis. Medline, Embase, Web of Science, and Scopus were searched for two groups of keywords related to 'biomarker' and 'SNHL'. Of the 7471 studies initially identified, 15 met the inclusion criteria. Based on the findings of these studies, biomarkers were grouped into six main categories based on their underlying pathological processes: heat shock proteins (HSP), immune-related, microRNA, neurotrophic, metabolome, and structural. For each category, this review identifies potential biomarkers that should be carefully validated in future studies. In particular, HSP70 and HSP90, complement components, mi1299, mi1270, and brain-derived neurotrophic factor regulated proteins may help direct future studies focused on characterising the biomarker profile in patients with SNHL. Future studies should ideally also utilise larger cohorts of patients with

**Data availability statement:** All relevant data generated or analysed during this study, including the minimal data set, are included in this published article.

**Funding:** Professor Bance and the SENSE Lab are supported by the NIHR Cambridge Biomedical Research Centre (NIHR203312*). The views expressed are those of the authors and not necessarily those of the NIHR or the Department of Health and Social Care. Sita Tarini Clark was funded by the Woolf Fisher Trust, New Zealand, the Cambridge Commonwealth, European, & International Trust, and by Trinity College, University of Cambridge. Nathan Creber was funded by the Garnett Passe and Rodney Williams Memorial Foundation.

**Competing interests:** The authors have declared that no competing interests exist.

specified hearing loss aetiologies. Such studies may help to pave the way towards a more accurate diagnosis of SNHL, improved prediction of cochlear implantation prognosis, and targeted therapeutic management.

---

## 1 Introduction

Disabling hearing loss is estimated to affect more than 430 million people worldwide and is projected to rise to 700 million by 2050 [1]. Despite this, our understanding of the aetiology of sensorineural hearing loss (SNHL) and the characteristics of the cochlear microenvironment remains limited. This is largely due to the live cochlea being a delicate and relatively inaccessible organ, due to the risk of loss of organ function upon tissue sampling, resulting in hearing and balance compromise. This reduced access has subsequently limited researcher's ability to investigate the pathogenesis and treatment of different types of SNHL, including Ménière's disease, vestibular schwannoma, presbycusis, and sudden idiopathic SNHL. Additionally, outcome prediction in patients undergoing cochlear implantation has remained difficult, with high inter-individual variability following surgery, even amongst patients with similar clinical histories and implanted devices [2].

Previous reviews of serum biomarkers have been informative in trying to further characterise the cochlear perilymph, but to a limited degree. Parham and colleagues first demonstrated that patients with BPPV have a significantly higher blood otolin-1 than patients without an otologic history [3]. Wang and colleagues also examined the diagnostic and predictive value of different serum metabolites within perilymph and found that N4-acetylcytidine, sphingosine, and nonadecanoic acid levels were all associated with hearing recovery in SNHL patients [4]. Lee et al. also identified multiple autoantibodies in patient serum that may be linked to an autoimmune reaction associated with bilateral SNHL [5]. However, these serum biomarkers have limited specificity and have therefore not been approved for use in clinical otology [6]. Indirect blood, serum, and urine samples are also limited by their lower biomarker concentrations and baseline variability in age, sex, and ethnicity, which further restricts the utility of their findings for diagnostic and therapeutic use in patients with SNHL [7–10].

To understand the pathophysiology of hearing loss it is therefore generally preferable to use tissues or samples in close proximity to the affected regions, as they are more likely to accurately represent the system studied. Recently, the establishment of a novel, safe, and feasible technique for intraoperative perilymph sampling has enabled the collection of a 'liquid biopsy' of the cochlea whilst still preserving inner ear function. Concerns regarding the safety of perilymph sampling have been addressed and the feasibility of perilymph sampling has been validated by several groups [11]. Perilymph liquid biopsies subsequently have the potential to more accurately characterise inner ear pathologies and provide clinicians with additional diagnostic tools to aid in the individualised treatment of SNHL. This is particularly useful as a prognostic indicator for performance after cochlear implantation and when considering potential therapeutic targets for other disorders.

Perilymph is an extracellular fluid in the inner ear that is found within the scala tympani and scala vestibuli of the cochlea, and in the vestibule of the inner ear [12]. It bathes the membranous labyrinth and functions to transmit sound vibrations from the oval window to the cochlear duct, as well as maintain the specific ionic components required to generate the endocochlear potential needed for transduction of mechanical to electrical energy [12]. The cochlea is a relatively closed fluid-filled system surrounded by a bony otic capsule. Due to the close proximity and contact of perilymph to the soft tissues structures within the scala tympani, it is reasonable to assume that a protein or metabolite secreted by damaged inner ear cells in pathological states will likely be found in higher concentrations in perilymph compared to cerebrospinal fluid (CSF) or blood [13]. Thus, direct perilymph sampling has significant potential to aid the identification of pathology-specific biomarkers in different inner ear diseases.

Initial attempts at postmortem human perilymph sampling helped to establish the ionic profile of perilymph [14,15]. Subsequent studies drew comparisons between CSF, serum, and perilymph proteins, including alpha-1 antitrypsin, pre-albumin, and esterase, to identify differences in the molecular profiles and hence, functions of the inner ear [16,17]. Animal studies have also demonstrated dynamic changes in perilymph composition that occur following noise exposure, including a four-fold increase in reactive oxygen species in mice after a 1-hour exposure to 110 dB [18]. While these studies have provided a foundation for our understanding of perilymph composition, they currently have limited clinical utility in the diagnosis of SNHL subtypes in human patients and the development of targeted treatments.

Perilymph samples can now be obtained during cochlear implantation, trans-labyrinthine resections of vestibular schwannomas, labyrinthectomy, and stapedotomy. This has enabled further characterisation of the inflammatory profile of different SNHL diagnoses and the comparison with perilymph samples from patients with conductive hearing loss (CHL), such as from patients with otosclerosis following stapedectomy [19]. Schmitt et al. have demonstrated the safety of these sampling methods by comparing pure-tone audiometry hearing thresholds in patients at the time of their first fitting of the cochlear implant and three months later [11]. The same stability in residual hearing was detected in patients who had undergone perilymph sampling as those that did not. No significant additional differences in hearing thresholds were found.

Lysaght and colleagues were the first to apply mass spectroscopy (MS) to human perilymph samples. They identified 71 proteins common in every sample obtained from twelve vestibular schwannoma patients [20]. As the cochlea is a tiny organ, only small volume samples between 1-5 µl can be safely obtained from the cochlear lumen without damaging function or other structures [21]. Therefore, subsequent advancements in MS-based protein analysis have focused on enhancing the sensitivity and reproducibility of protein identification within these smaller samples. More recent studies have successfully detected approximately 300 different proteins within a single perilymph sample of 1-12 µl [13,22]. Such analytical advances, combined with the establishment of safe and sensitive methods of obtaining a 'liquid biopsy' of the inner ear, have provided valuable insight into the previously uncharted microenvironment of the cochlea.

The complexity of the cochlea microenvironment has implicated roles for the proteome [11], metabolome [23], miRNA [24], and inflammasome [25], in the pathogenesis of SNHL. The primary aim of this review is to identify biomarkers from perilymph samples associated with SNHL to inform the diagnosis of specific pathologies. The secondary aims of this review are to investigate biomarkers associated with cochlear implantation outcomes to help predict patient prognosis and identify potential future therapeutic targets.

## 2 Materials and methods

### 2.1 Search strategy and databases

A search was conducted on the following databases after consulting a specialist librarian (VP): Medline (via Ovid), Embase (via Ovid), Web of Science (Core Collection), and Scopus from inception to April 2024. The search strategy was peer-reviewed by two librarian colleagues of VP using the Peer Review of Electronic Search Strategies (PRESS)

checklist [26], and evaluated against the PRISMA-S guidelines [27]. Databases were searched separately rather than multiple databases being searched on the same platform. The search syntax was adapted for each database, to account for variation in thesaurus terms/controlled vocabulary between databases. In the preliminary stage of this study, Google Scholar was used to identify a broad range of target papers that otherwise may not have been captured by conventional databases. Subsequently, a formal search strategy was performed using only established databases, as listed above.

Keywords were divided into two groups according to biomarker type and hearing loss pathologies. The first group included: Biomarker, Metabolome, Proteome, RNA, miRNA, or Inflammasome, and the second group included: Hearing loss, Deafness, Ménière, or Otosclerosis. Excluding animal studies resulted in a minority of the target papers being omitted as not all human studies were tagged appropriately. Therefore, animal studies were included in the primary search and subsequently removed at screening. This was achieved by excluding studies if the population described was clearly animal-based (e.g. rodent, primate, or other non-human models). Where abstracts were unclear, full texts were retrieved to assess the study population; this was performed by two independent reviewers. This process ensured that only human studies relevant to the review question were retained in the final analysis.

## 2.2 Study eligibility, inclusion, and exclusion criteria

Inclusion criteria for this study included: peer-reviewed studies; biomarkers from perilymph associated with hearing loss; human studies; international publications; and correlational studies. Correlational studies included cohort studies in which participants were grouped into different SNHL aetiologies with or without a control group, and associated perilymph biomarkers were identified. Exclusion criteria included: animal studies; case studies; inner ear tissue biopsy/cellular samples; endolymphatic sac fluid; review articles; vertigo; tinnitus; serum biomarker; genetic markers; conference poster articles; non-peer-reviewed studies. Only primary studies were included in this systematic review. Review articles were excluded to avoid double-counting evidence and introducing bias into the study conclusions. Similarly, case studies were also excluded due to their limited generalisability.

## 2.3 Study screening process

After duplicate removal, 7471 papers were identified. The title and abstract screening were completed by two independent reviewers (KW and NC) using the online platform, Rayyan. The most common reasons for exclusion at the title and abstract screening stage were: patients with other types of hearing loss (non-SNHL), reviews, and commentaries. 191 studies were subsequently selected for full text screening. At this stage, the most common reason for exclusion was different sample type (typically serum, CSF, or inner ear tissue). Previously, studies were reliant on these indirect systemic sample types when investigating the cochlear microenvironment. However, due to recent advances in safe, intraoperative sampling techniques, researchers are now able to include direct perilymph samples in their analysis. In turn, this review focused on the biomarker profile of these localised perilymph samples specifically, to help inform future studies utilising this novel technique in different hearing pathologies going forward. 14 eligible papers were identified from this screening. Finally, the reference list of each of the remaining papers were screened, and one additional paper was identified. A total of 15 papers were therefore included in this review, as outlined in (Fig 1).

## 2.4 Data extraction and analysis

The extracted data included: author, year of publication, country of publication, study design, level of evidence according to 'The Oxford Centre of Evidence-Based Medicine', perilymph extraction procedure (e.g., cochlear implantation/ vestibular schwannoma resection/ stapedectomy), biomarker identified, and key findings. Some studies included a large volume of screened biomarkers. Those that produced statistically significant results were included in the study analysis.

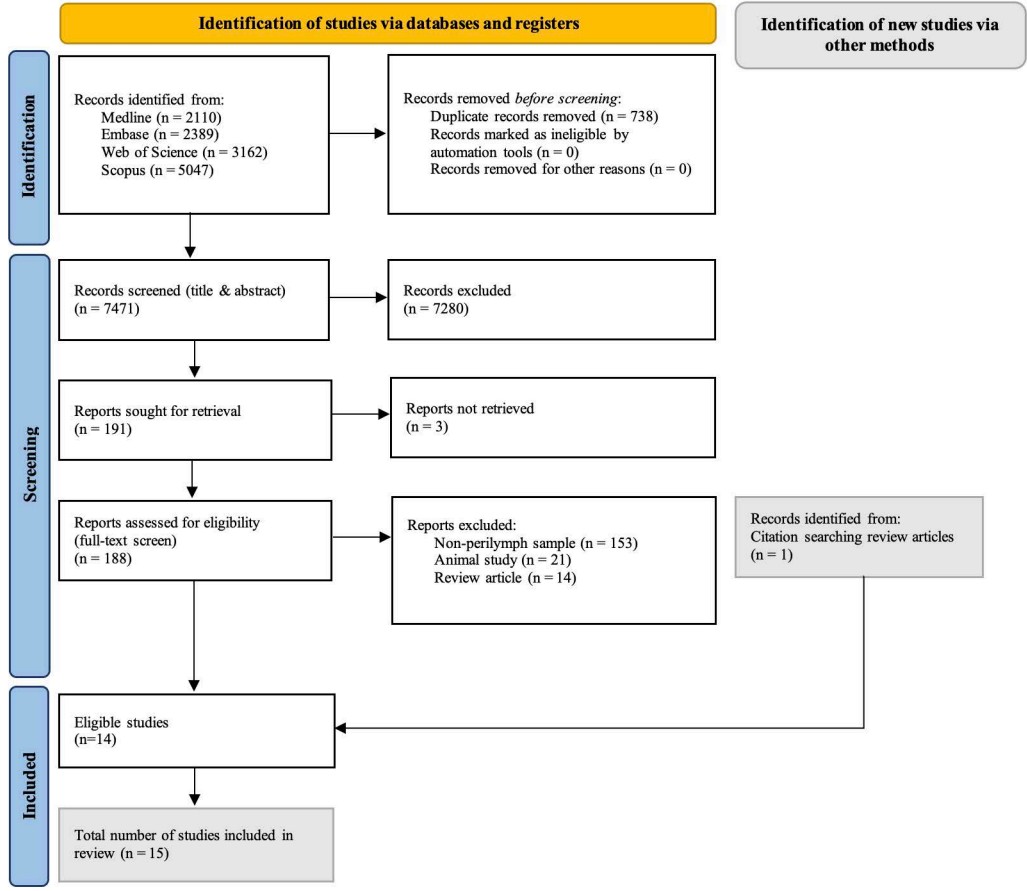

**Fig 1. PRISMA flow diagram.**

## 2.5 Risk of bias

The ROBINS-I tool was used to assess the risk of bias in the included studies. This tool evaluates seven domains of potential bias: confounding, selection of participants, classification of interventions, deviations from intended interventions, missing data, measurement of outcomes, and selection of reported resources. Each domain was assessed as low, moderate, or series risk of bias. The risk of bias was assessed by two independent reviewers (KJ and SC), and discrepancies were resolved through discussion or consulting a third reviewer (NC).

## 3  Results

All 15 studies included in this review were correlational in design, as outlined in Table 1 [11,20,22,24,25,28–37]. One study utilised machine learning approaches to build disease-specific algorithms to predict the presence of SNHL using only miRNA expression profiles [34]. The majority of studies sampled perilymph in patients with SNHL via the round window during cochlear implantation. In studies with control groups, perilymph was typically obtained from patients with CHL due to otosclerosis during stapedectomy [29,34–36], labyrinthectomy for Ménière's disease [35], or trans-labyrinthine surgery for vestibular schwannoma resection [11,20,30–32]. Beyond differences in sampling location, the aetiology of SNHL varied between studies. Most patients in these cohorts had SNHL due to Ménière's disease, vestibular schwannoma, or an enlarged vestibular aqueduct. In nine of the studies, the aetiology of SNHL was not reported for several patients. One study had an unspecified sample size [24].

**Table 1. Summary of published perilymph biomarkers associated with hearing loss or cochlear implant outcome.**

| Author, Year | Aetiology of Hearing Loss | Key Biomarkers | Key Findings |
|---|---|---|---|
| Arambula et al., 2023 [28] | SNHL unknown aetiology (n = 26)<br>Ménière's disease (n = 12)<br>Otosclerosis (n = 10)<br>Vestibular schwannoma (n = 10)<br>Enlarged vestibular aqueduct (n = 14) | Filamin-B<br>ACTB<br>CLIC1<br>GSTP1 | Distinct clustering of Ménière's disease perilymph samples from patients with other aetiologies of hearing loss. These 2 structural proteins were highly abundant. |
| Ciorba et al., 2010 [29] | SNHL unknown aetiology (n = 49)<br>Genetic (n = 33)<br>Infective (n = 7)<br>Otosclerosis (n = 7)<br>Cerebral palsy (n = 3) | ROS | Elevated ROS levels were detected in the perilymph of patients with profound SNHL. This was attributed to the activity of the xanthine dehydrogenase/xanthine oxidase enzyme system. |
| de Vries et al., 2019 [30] | SNHL of varying aetiology (n = 38)<br>Vestibular schwannoma (n = 4)<br>Normal hearing (n = 3) | BDNF-regulated proteins<br>PLTP | Patients with profound hearing loss had decreased expression of specific BDNF-regulated proteins compared to patients with some residual hearing following cochlear implantation. PLTP showed a statistically significant correlation to preoperative hearing thresholds. |
| Durisin et al., 2022 [22] | Unknown (n = 14)<br>Not specified (n = 9)<br>Ménière's disease (n = 7)<br>Otosclerosis (n = 7)<br>Enlarged vestibular aqueduct (n = 5)<br>Cytomegalovirus infection (n = 1)<br>Rubella embryopathy (n = 1)<br>Meningitis (n = 1) | HSPA1B<br>Immunoglobulins<br>Myeloperoxidase<br>C8A<br>Attractin<br>SELENBP1<br>Plasma kallikrein | 5 proteins were identified to be in significantly higher abundance in the excellent hearing group following cochlear implantation. 6 proteins were identified to be in significantly higher abundance in the poor hearing group following cochlear implantation. |
| Edvardsson Rasmussen et al., 2018 [31] | Vestibular schwannoma (n = 15) | Alpha-2-HS glycoprotein | Alpha-2-HS glycoprotein was found to be an independent variable for tumour-associated hearing loss. |
| Lysaght et al., 2011 [20] | Vestibular schwannoma (n = 12) | μ-crystallin<br>LRP2 | 15 candidate biomarkers for vestibular schwannoma were associated with poor hearing. μ-crystallin and LRP2 were identified in both vestibular schwannoma samples, but were not present in CI samples. |
| Schmitt et al., 2017 [11] | Unknown (n = 27)<br>Vestibular schwannoma (n = 4)<br>Ménière's disease (n = 2)<br>Cytomegalovirus infection (n = 1)<br>Enlarged vestibular aqueduct (n = 1)<br>CHARGE syndrome (n = 1)<br>Meningitis (n = 1)<br>Auditory neuropathy (n = 1) | ß-Ala-His dipeptidase<br>Dickkopf-related protein 3<br>CD14<br>Immunoglobulin α-chains<br>Complement C4-ß chain<br>Angiotensinogen | Some proteins were only found in adults and may be correlated with the presence of presbycusis. 12 proteins showed higher abundance in the adult group compared to children. |
| Schmitt et al., 2018 [32] | Unknown (n = 27)<br>Vestibular schwannoma (n = 4)<br>Ménière's disease (n = 2)<br>Cytomegalovirus infection (n = 1)<br>Enlarged vestibular aqueduct (n = 1)<br>CHARGE syndrome (n = 1)<br>Meningitis (n = 1)<br>Auditory neuropathy (n = 1) | HSP 90<br>HSP 70 (subtype 1 and 6) | Same dataset from the study above. HSP90 correlated with a complete loss of residual hearing group following cochlear implantation. HSP70 Subtype 1 and 6 were both identified in all patients with hearing preservation following cochlear implantation. No significant difference in HSP distribution in the vestibular schwannoma group. |
| Schmitt et al., 2021 [33] | Ménière's disease (n = 12)<br>Enlarged vestibular aqueduct (n = 10)<br>Otosclerosis (n = 9) | SDR9C7<br>ESD | 33 proteins found in higher abundance in Ménière's disease patients compared to control groups. SDR9C7 and ESD were uniquely identified in the perilymph of Ménière's patients. |
| Shew et al., 2018 [24] | Ménière's disease (n = 2)<br>Unknown (number & aetiology of controls not specified) | mi1233-5p<br>mi455-3p<br>mi638 | miRNAs involved in regulating two proteins Aquaporin 4 and TNFSF1 were correlated with Ménière's disease and not found in control samples. |

*(Continued)*

 

**Table 1.** (Continued)

| Author, Year | Aetiology of Hearing Loss | Key Biomarkers | Key Findings |
|---|---|---|---|
| Shew et al., 2019 [34] | Unknown (n = 12)<br>Otosclerosis (n = 4) | miR-184<br>miR-660<br>miR-Let 7a 5p<br>miR-3142<br>miR-335 | The Let 7 miRNA family were critical in differentiating between CI patients with and without residual hearing following surgery. |
| Shew et al., 2021 [35] | Ménière's disease (n = 5)<br>Otosclerosis (n = 5) | miR-1299<br>miR-1270<br>miR-3960<br>miR-4481<br>miR-675 | Identified 16 differentially expressed miRNA in Ménière's disease samples, 6 of them related to aquaporin expression and 12 in autoimmune, inflammatory. miRNA-1299 was found exclusively in the Ménière's disease samples. |
| Shew et al., 2021 [36] | SNHL unknown aetiology (n = 14)<br>Otosclerosis (n = 4) | miR-1207<br>miR-4651 | Lower expression of miR-1207 and miR-4651 was statistically correlated with poorer pure tone assessment in patients undergoing CI surgery. There was no variation in miRNA expression between otosclerosis patients. |
| Trinh et al., 2019 [37] | Congenital (n = 8)<br>Presbycusis (n = 5)<br>Sudden hearing loss (n = 2)<br>Temporal bone fracture (n = 2)<br>Ménière's disease (n = 1)<br>Trauma (n = 1) | N-acetylneuraminate<br>Glutaric acid<br>Cystine<br>2-methylpropanoate<br>Butanoate<br>Xanthine | Profiles were different for patients with over 12 years of hearing loss compared to less than 12 years. Identified a correlation between levels of N-acetylneuraminate and duration of hearing loss. |
| Warnecke et al., 2019 [25] | SNHL aetiology unknown (n = 15)<br>Ménière's disease (n = 2)<br>Meningitis (n = 2)<br>Vestibular schwannoma (n = 1)<br>Congenital (n = 14) | VEGF-D<br>IGFBP1<br>IL-13<br>IL-9 | VEGF-D was reduced levels in patients with complete deafness prior to cochlear implantation. Higher IGFBP1 in patients with complete hearing loss. Higher concentrations of IL-13 and IL-9 in complete loss compared to residual hearing before cochlear implantation. |

Key: SNHL = Sensorineural Hearing Loss; ACTB = Actin, Cytoplasmic 1; CLIC1 = Chloride Intracellular Channel Protein 1; GSTP1 = Glutathione S-transferase P; ROS = Reactive Oxygen Species; BDNF = Brain Derived Neurotrophic Factor; PLTP = Phospholipid Transfer Protein; HSPA1B = Heat Shock 70 kDa Protein 1A/B; C8A = Complement Component C8 Alpha Chain; SELENBP1 = Selenium-Binding Protein 1; Alpha-2-HS Glycoprotein = Alpha-2 Heremans Schmid Glycoprotein; LRP2 = Low Density Lipoprotein Receptor-related Protein 2; CI = Cochlear Implant; CD14 = Cluster of Differentiation 14; HSP = Heat Shock Protein; SDR9C7 = Short-chain Dehydrogenase/Reductase Family 9C member 7; ESD = S-formylglutathione hydrolase; miRNAs = microRNAs; VEGF-D = Vascular Endothelial Growth Factor D; IGFBP1 = Insulin-like Growth Factor Binding Protein 1; IL = Interleukin.

Two predominant study designs were identified in this review. Firstly, five studies sought to identify biomarkers that correlated with cochlear implant outcomes, particularly loss of residual hearing [22,30,32,34,36]. This involved performing pure tone audiometry following cochlear implantation to group patients into those with residual hearing and those without any residual hearing. The biomarker profile was compared between the two groups. One study grouped patients into "complete deafness" and "residual hearing" before cochlear implantation [25]. The second predominant study design aimed to identify biomarkers correlated with specific SNHL aetiologies. Five studies focused on biomarkers associated with Ménière's disease in comparison to otosclerosis controls [24,28,33,36]. Two studies looked at biomarkers associated with vestibular schwannomas [20,31]. Interestingly, one study compared biomarker profiles based on the duration of hearing loss (more than 12 years compared to less than 12 years) [37]. Another study compared the biomarker profile of adults and children undergoing cochlear implantation in the hope of gaining potential insights into the pathogenesis of presbycusis [11]. In the following discussion, the findings are discussed by functional group in more detail.

Fourteen studies were classified as moderate bias risk and one study as severe bias risk (Table 2). Importantly, most of the studies were deemed 'moderate' in the 'bias due to confounding' domain, due to the inclusion of participants with unknown hearing loss aetiologies. One study did not specify the number of patients involved or any further disease aetiology [24]. In some of the studies, bilateral samples were taken from patients [11,22,25,30] but were analysed separately. One paper adopted an unusual methodology for sample analysis; twelve patients were included in the study, six with

**Table 2. Risk of bias analysis of included studies using the ROBINS-I quality assessment tool.**

| | | Risk of Bias Domains | | | | | | | |
|---|---|---|---|---|---|---|---|---|---|
| | | D1 | D2 | D3 | D4 | D5 | D6 | D7 | Overall |
| Author, Year | Arambula et al., 2023 [28] | − | + | + | + | + | + | + | − |
| | Ciorba et al., 2010 [29] | − | − | + | + | − | + | + | − |
| | de Vries et al., 2019 [30] | − | − | + | + | − | + | + | − |
| | Dursin et al., 2022 [22] | − | + | + | + | − | + | + | − |
| | Edvardsson Rasmussen et al., 2018 [31] | − | + | + | + | − | + | + | − |
| | Lysaght et al., 2011 [20] | − | − | + | + | − | + | + | − |
| | Schmitt et al., 2017 [11] | − | − | + | + | − | + | + | − |
| | Schmitt et al., 2018 [32] | − | − | + | + | − | + | + | − |
| | Schmitt et al., 2021 [33] | − | − | + | + | + | + | + | − |
| | Shew et al., 2018 [24] | ! | ! | + | + | ! | + | + | ! |
| | Shew et al., 2019 [34] | − | − | + | + | − | + | + | − |
| | Shew et al., 2021 [35] | − | + | + | + | − | + | + | − |
| | Shew et al, 2021 [36] | − | + | + | + | + | + | + | − |
| | Trinh et al., 2019 [37] | − | + | + | + | + | + | + | − |
| | Warnecke et al., 2019 [25] | − | + | + | + | + | + | + | − |

Domains:
D1: bias due to confounding
D2: bias due to selection of participants
D3: bias due to classification of events
D4: bias due to deviations from intended interventions
D5: bias due to missing data
D6: bias in the measurement of outcome
D7: bias the selection of reported results

Judgement:

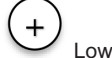 Low risk

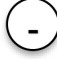 Moderate risk

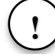 Severe risk

vestibular schwannomas and six post-CI; however, only four samples were run for analysis and samples were pooled to result in more protein for analysis [20].

## 4 Discussion

Previous reviews have adopted varying approaches to biomarker classification. For example, Gomaa et al. categorised the biomarkers identified in their study according to their diagnostic, predictive, and prognostic value [6]. Appraising the functional diversity of perilymph biomarkers identified through this review, we have grouped biomarkers into the following functional categories: heat shock proteins, immune/inflammatory mediators, microRNAs, neurotrophic pathway, metabolic, and structural. The rationale for this categorisation is that it may help to discern the key pathological processes underlying SNHL within a complex perilymph microenvironment.

### 4.1 Heat Shock Proteins

Heat shock proteins (HSP) are a superfamily of proteins that play an important role in cellular protection, particularly under stress conditions such as heat, oxidative stress, or toxins [38]. Mechanical and chemical stressors can result in various changes in genomic transcription and protein folding. In response, however, HSPs work to protect cells from injury by promoting the folding of denatured proteins and preventing aggregation. Moreover, HSPs can suppress apoptotic pathways by interacting with proteins associated with signal transduction in active cell death [39]. These proteins are classified according to their weight into 6 families: small HSPCs, HSP40, HSP60, HSP70, HSP90, and HSP110 [40]. The neuroprotective role of HSPs have also been implicated beyond the cochlea, with HSP70 and small HSPs associated with enhanced retinal ganglion cell survival in optic neuropathies such as glaucoma [39].

In the cochlea, HSPs may play a protective role in hearing preservation following cochlear implantation. The variability in outcomes of cochlear implant patients remains largely unexplained. Clinical factors alone have proved insufficient for predicting patient performance [41], suggesting that additional factors such as the patient's genetic profile and their individual cochlear microenvironment may influence their postoperative outcomes, particularly with regards to hearing preservation following cochlear implantation. Schmitt et al. evaluated the HSPs associated with residual hearing after cochlear implantation and found that subtypes 1 and 6 of HSP70 were found in all patients with preserved residual hearing [32]. The protective effect of HSP70 is supported by experimental mouse models in which HSP70-overexpressing mice were significantly protected against aminoglycoside-induced hearing loss and hair cell death compared to their wild-type counterparts [42].

Interestingly, not all HSPs exert a protective effect. HSP90 was found in higher levels in patients with a complete loss of residual hearing following cochlear implantation and only one patient with preserved hearing had HSP90 detected in the perilymph following surgery. In this study, 3 patients undergoing translabyrinthine vestibular surgeries were also included, however, no significant difference in HSP distribution was identified when compared to the cochlear implantation group. Furthermore, Edvardsson Rasmussen et al. identified three variations of HSP70 in only one of the 15 patients with vestibular schwannoma included in their study [31].

Of note, Schmitt et al.'s [32] findings were not replicated in Durisin et al.'s [22] analysis of perilymph proteins post-CI. These researchers evaluated hearing performance using speech intelligibility (HSM sentence test in noise at 10dB and Freiberg monosyllable word test) and grouped participants into excellent and poor speech intelligibility groups post-CI. They found higher levels of HSP70 1A/B in the poor speech intelligibility group, in other words, higher levels of this HSP in the group with poorer CI outcomes. Overall, the association of HSP70 with CI outcomes is inconsistent, with further investigation needed to clarify the role of HSPs as a useful marker for CI prognostication.

### 4.2 Immune/ Inflammatory mediators

Historically, the inner ear, including the cochlea, was assumed to be an immune-privileged organ similar to the eyes or brain, due to the presence of a blood-labyrinthine-barrier (BLB) with tight junctions [43]. This forms a protective barrier

blocking immune cells, inner ear antigens, and antibodies from entering. However, recent studies have demonstrated the presence of a complex immune microenvironment within the cochlea itself [44]. An initial study demonstrating the responsiveness of certain types of SSHL to steroid treatment was the first to challenge the idea of the cochlea as an immune-privileged organ [45]. Later, Zhang and colleagues revealed two cell types of macrophage lineage in the lateral wall of the cochlea: cochlear macrophages and perivascular macrophage-like melanocytes, which are involved in clearing cell debris, maintaining cochlear fluid homeostasis, and regulating BLB permeability [46]. Animal studies have demonstrated changes in inflammatory gene expression in the cochlea following noise exposure, including the upregulation of genes such as CXL10, SOCS3, and TCl1b1 [47]. Other studies have identified increased pro-inflammatory cytokines such as TNF-α, IL-1β, and IL-6 following noise exposure [48]. Since these initial findings, an extensive list of inflammatory gene regulations has been reported by Karayay and colleagues [49]. Collectively, these findings suggest a role for inflammatory cells/mediators in the development of cochlear damage and SNHL.

Soluble components of the innate immune system have recently been identified in the perilymph which correlate with cochlear implant performance. Warnecke and colleagues identified higher concentrations of inflammatory cytokines, IL-13 and IL-9, in patients with complete hearing loss following CI surgery when compared to patients with some residual hearing following surgery, although the difference in IL-9 did not reach statistical significance [25]. The complement system is another crucial part of the innate immune system and has roles in pathogen opsonisation, chemotaxis and activation of leukocytes, cytolysis, as well as clearance of apoptotic cells and immune complexes [50]. Higher levels of complement component C8 alpha chain were found in patients with excellent speech intelligibility at 1-year post-CI compared to the poor speech intelligibility group [22]. Complement C8 is an important antibacterial immune effector and is part of the membrane attack complex that forms a pore in bacterial membranes and leads to cell lysis. Complement C1r subcomponent and complement H were also found to be in higher abundance in the perilymph of patients with Ménière's disease compared to the perilymph of otosclerosis controls [33]. Furthermore, complement C4 was found in a higher abundance in adults when compared to children following cochlear implantation, potentially providing insight into differences in perilymph composition in presbycusis by age [11]. Finally, one study identified alpha-2-HS glycoprotein as an independent variable for tumour-associated hearing loss [31]. Alpha-2-HS glycoprotein is an acute phase response protein and is involved in neutrophil degranulation [51]. Recent studies have also hypothesised that alpha-2-HS may be excreted from vestibular schwannomas into the extracellular perilymph where it exerts pro-inflammatory activity [31]. These findings are in congruence with a previous study by Schmitt et al. that identified alpha-2-HS in several samples from patients with an unknown cause of sensorineural hearing loss [11].

Components of the adaptive immune system, specifically immunoglobulin (Ig) chains, have also been associated with SNHL. One study found Ig Kappa chain V-IV regions upregulated in patients with excellent hearing performance following cochlear implantation. However, a different subset of immunoglobulins was upregulated in patients with poor hearing performance, specifically genes IGHV1–2 and IGHV1–46 which encode various regions on the immunoglobulin heavy chain, and IGKV6–21 for immunoglobulin kappa variable 6–21 [22]. Two further studies have also suggested a role for attractin in SSHL [11,22]. Attractin is a circulating glycoprotein that is rapidly expressed on activated T cells [52] and was found to be more abundant in patients with Ménière's disease but also patients with higher speech intelligibility after CI surgery. Collectively, these findings suggest that a diverse set of immune or inflammatory mediators may be associated with SNHL, with a potential impact on CI performance, ranging from interleukins and complement proteins to acute phase reactants and immunoglobulins.

### 4.3 microRNAs

Changes in the protein environment, or 'proteome' of the inner ear may also contribute to SNHL, a process regulated in part by microRNAs (miRNA). MiRNAs are 19–23 base pair single-stranded RNA sequences that regulate gene expression through mRNA degradation and silencing [53,54]. Beyond the ear, miRNAs in CSF are being evaluated as potential

markers of neurodegenerative diseases such as Alzheimer's and Parkinson's [55] and in the aqueous humour of the eye for glaucoma [56]. Equally, miRNAs have been shown to play a crucial role in inner ear development. The miR-183 family is expressed as the otic vesicle develops from the otic placode and is later restricted in expression to hair cells and the spiral ganglion [57].

Previous studies on peripheral blood samples identified 24 differentially expressed miRNAs in SNHL patients when compared to healthy controls, with subsequent functional annotation analysis revealing target genes in arachidonic acid metabolism, complement, and coagulation cascades [58]. Focusing on perilymph samples in particular, work by Shew et al. [24,34–36] has identified a diverse set of miRNAs correlated with SNHL of varying aetiologies. In their 2018 study, three miRNAs (mi1233-5p, mi455-3p, mi638) were expressed in patients undergoing cochlear implantation who were diagnosed with Ménière's disease, but not in the other implantation patients without Ménière's disease [24]. Mi1233-5p, mi455-3p, and mi638 form part of a regulatory network, including two proteins, aquaporin 4, and TNFSF12, which have been implicated in the pathogenesis of Ménière's disease [59,60]. However, caution should be exercised when interpreting these findings, as the number of control participants was not stated in this study [24]. Shew and colleagues subsequent study in 2021 corroborated these findings and identified miRNA 1299 and 1270 uniquely and differentially expressed in the perilymph of patients with Ménière's disease [35]. Both miRNA 1299 and 1270 are linked to inflammatory and autoimmune pathways, while miRNA-1299 can be linked to aquaporin expression specifically. The author's 2019 study also utilised a machine learning approach and identified several 'critical miRNA' that differentiated conductive and SNHL. The downstream targets were identified as KCNJ10, HCN, and Otoferlin which are important for propagating endocochlear potentials [61], post-synaptic potentials [62], and Ca2+ evoked vesicular endocytosis in inner hair cells [63], respectively.

In addition to aquaporin and ion channel expression, miRNAs may also play a role in the neurotrophin pathway as discussed below. Shew and colleagues found that a lower expression of miR-1207 and miR4651, which target neurotrophin receptors 2 and 3 (NTR2 and NTR3), was statistically correlated with having no residual hearing (pure tone audiometry threshold > 80dB) prior to cochlear implantation [35]. These findings demonstrate the potential clinical utility of implementing miRNA profiling before CI surgery, to optimise post-operative outcomes in patients with SNHL.

### 4.4 Neurotrophin pathway proteins

As previously suggested, neurotrophic proteins may represent an important group of perilymph biomarkers for SNHL with several studies directly investigating their presence in the perilymph. Neurotrophins are a family of molecules that play an integral role in the development and support of neurons. In the murine inner ear, BDNF and NT-3 have been shown to regulate the connection of hair cells to neurons [5,64]. One study found that BDNF-regulated proteins were correlated with residual hearing prior to cochlear implantation and improved performance after 1 year, although the samples were obtained from patients with SNHL of various aetiologies [25]. Another study found higher levels of kallikrein in patients with excellent hearing post-CI surgery [22] which works to increase the expression of BDNF and pro-survival Bcl-2 genes [65]. However, Schmitt and colleagues did not detect endogenous BDNF in their samples [11].

Endothelial factors may also play a role in inner ear repair. Warnecke and colleagues found reduced levels of VEGF-D in patients with no residual hearing (PTA threshold > 80dB) compared to patients with residual hearing following cochlear implantation [25]. Interestingly, VEGF-D has a central role in lymphangiogenesis [66], suggesting that reduced levels of VEGF-D in patients with no residual hearing may indicate ongoing damage and reduced repair. Similarly, Schmitt and colleagues found higher levels of HSPG in patients with enlarged vestibular aqueducts and Ménière's disease, compared to those with otosclerosis [33]. HSPG has complex regulatory roles over VEGF A, VEGFR2, and α2β1 integrin signalling axes [67]. Another growth factor regulator, insulin-like growth factor binding protein 1 (IGFBP1), which regulates insulin-like growth factor 1 (IGF-1), was higher in patients with no residual hearing [25]. Although, the precise correlation of neurotrophins and vascular factors with SNHL is unclear, they are likely to provide insight into the pathogenesis of SNHL and the aetiologies of related diseases.

## 4.5 Metabolome

A growing body of literature suggests that the metabolic profile of perilymph may provide insight into SNHL pathogenesis. Trinh and colleagues found a correlation between the metabolic profile of perilymph and the duration of hearing loss in patients (less than 12, or more than 12 years of hearing loss), specifically elevated levels of N-acetylneuraminate, which is a metabolite located on the terminal glycoprotein and glycolipid on the surface of cell membranes that increases following cell membrane breakage (hair cell apoptosis) [37]. Other discriminant metabolites found in this study include: glutaric acid, cysteine, butanoate, and xanthine. Similarly, another study found the protein short-chain dehydrogenase/reductase family 9C member 7 (SDR9C7) uniquely in 11 out of 12 samples from Ménière's disease patients, but not in patients with an enlarged vestibular aqueduct or otosclerosis [33]. SDR9C7 is involved in vitamin A metabolism, specifically the conversion of retinal to retinol in the presence of NADH. Mutations in SDR9C7 have already been linked to dermatological pathology (autosomal recessive ichthyosis). However, the role of this enzyme in the context of the cochlea is yet to be explored.

Reactive oxygen species (ROS) also form a component of the metabolic profile of perilymph. ROS are responsible for direct cellular damage to lipids, proteins, and DNA, and trigger apoptosis or necrosis [68]. Elevated levels of ROS have been detected in patients with profound SNHL and have been attributed to the activity of the xanthine dehydrogenase/xanthine oxidase enzyme system [29]. One study found that proteins related to apoptotic signalling and ROS were upregulated following intratympanic steroid treatment, specifically serine leukocyte inhibitor and annexin A1 which are both involved in ROS processes and have anti-inflammatory activities [69]. In this way, several metabolic markers could provide insight into the pathogenesis of SNHL.

## 4.6 Structural proteins

Finally, a selection of structural proteins has been detected at higher levels in SNHL patients than in controls. Flnb and ACTB, which encode filamin B and beta-actin respectively, were found to be highly abundant in Ménière's disease patients compared to those with alternative hearing loss aetiologies [28]. Six out of the seven highly abundant proteins in Ménière's disease found by Arambula and colleagues [28] were also identified in Schmitt and colleagues' [33] list of 33 proteins. Another study reported low-density lipoprotein-related protein 2 (LPR2) in the perilymph of vestibular schwannoma patients but not controls undergoing CI without vestibular schwannoma [20]. LRP2 is a transmembrane receptor protein found primarily in absorptive epithelial cells and is a key player in mediating endocytosis and mutations in LPR2 are associated with Donnai-Barrow syndrome and facio-oculo-acoustico-renal syndrome, which present with SNHL [70]. These structural proteins present potential candidates for perilymph biomarkers and should be carefully investigated in future work.

## 5 Future directions

Taken together, a diverse group of biomarkers have been identified in human perilymph and correlated with SNHL pathologies of varying aetiology, as demonstrated in (Fig 2). This systematic review has identified six main groups of biomarkers that may inform future clinical studies on cochlear sampling. Future targeted studies with larger cohorts and clearer diagnoses are required to help further characterise the prevalence and functional role of these biomarkers. This review highlighted two main study designs: those that compared biomarker profiles based on cochlear implant outcomes, and those that characterised a specific SNHL aetiology by comparison to otosclerosis controls. The authors recommend these methodologies be adopted in future studies to facilitate ongoing comparison of results between studies. Although, one limitation of sampling at the time of cochlear implantation is that it may not fully capture the range of pathogenic mechanisms involved earlier in disease and those in response to the implant following CI surgery.

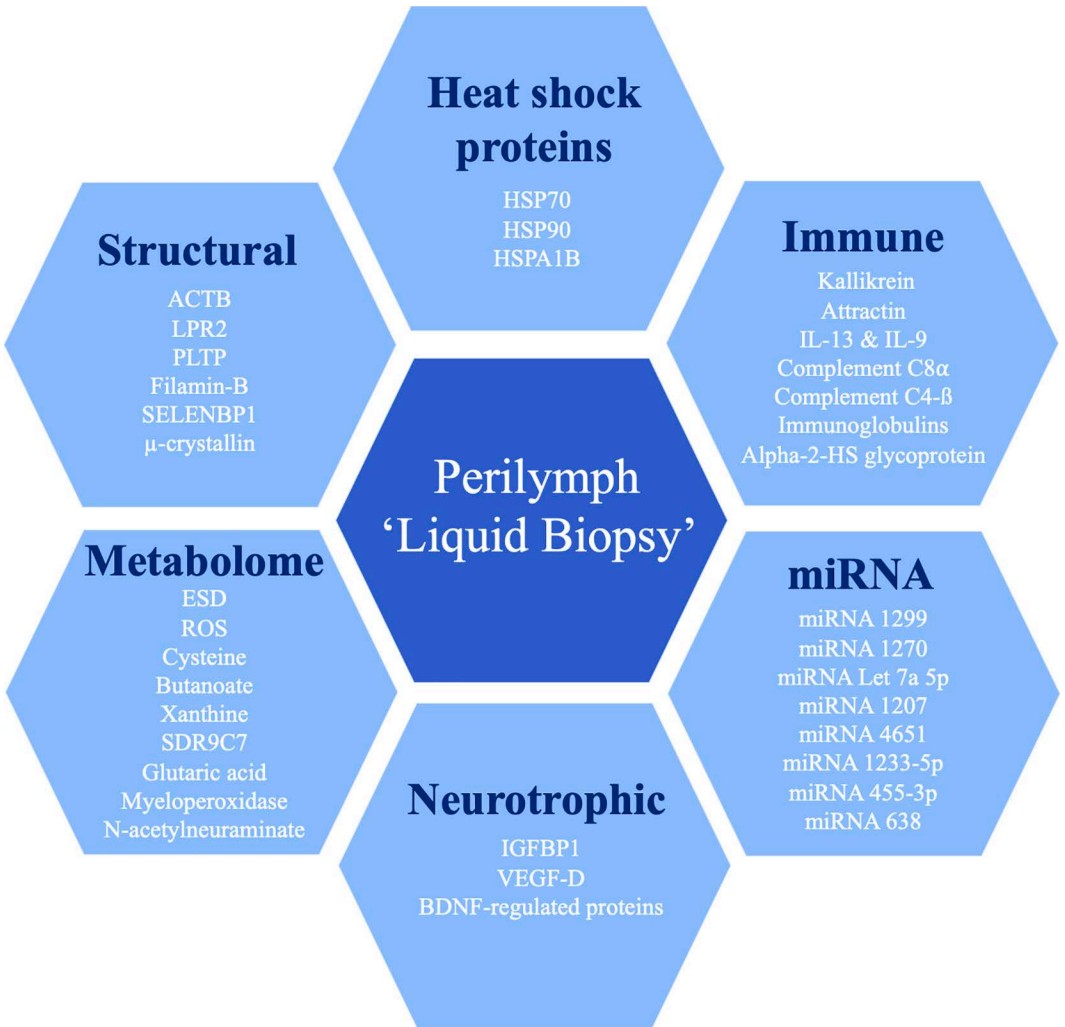

**Fig 2. Schematic of perilymph 'liquid biopsy' biomarkers grouped into functional categories.** Key: HSP = Heat Shock Protein; HSPA1B = Heat Shock 70kDa Protein 1A/B; ACTB = Actin Beta, Cytoplasmic 1; LRP2 = Low Density Lipoprotein Receptor-related Protein 2; PLTP = Phospholipid Transfer Protein; SELENBP1 = Selenium-Binding Protein 1; IL = Interleukin; Alpha-2-HS glycoprotein = Alpha-2 Heremans Schmid Glycoprotein; ESD = S-Formylglutathione Hydrolase; ROS = Reactive Oxygen Species; SDR9C7 = Short-chain Dehydrogenase/Reductase Family 9C member 7; miRNAs = microRNAs; IGFBP1 = Insulin-like growth factor binding protein 1; VEGF-D = Vascular Endothelial Growth Factor D; BDNF = Brain Derived Neurotrophic Factor.

## 6 Conclusion

Analysis of human perilymph has provided novel insights into the complex cochlear microenvironment and shows promise for expanding our understanding of the pathophysiology and ability to treat SNHL. The cochlear microenvironment is evidently more complex than first recognised with roles for heat shock proteins, neurotrophin factors, inflammatory mediators, structural proteins, and metabolites as outlined in this review. The diversity of biomarkers identified, coupled with the heterogenous aetiologies of SNHL, complicates researcher's abilities to identify direct correlations between biomarkers and hearing loss. Standardisation of methodology and larger controlled studies will be required to establish reliable and translatable results. Overall, sampling human perilymph is a safe procedure that provides insight into a previously 'black box' organ and could help characterise and advance the treatment of SNHL.

## Acknowledgments

The authors are very grateful to Veronica Phillips (Cambridge University Medical Librarian) for her help in creating and running the search for this systematic review.

## Author contributions

**Conceptualization:** Neil Donnelly, Manohar Bance, Nathan Creber.

**Data curation:** Kujani Wanniarachchi, Neil Donnelly, Manohar Bance, Nathan Creber.

**Formal analysis:** Kujani Wanniarachchi, Sita Tarini Clark, Manohar Bance, Nathan Creber.

**Funding acquisition:** Sita Tarini Clark, Manohar Bance, Nathan Creber.

**Investigation:** Kujani Wanniarachchi, Sita Tarini Clark, Neil Donnelly, Nathan Creber.

**Methodology:** Kujani Wanniarachchi, Sita Tarini Clark, Manohar Bance, Nathan Creber.

**Project administration:** Neil Donnelly, Manohar Bance, Nathan Creber.

**Resources:** Kujani Wanniarachchi, Sita Tarini Clark, Manohar Bance, Nathan Creber.

**Software:** Kujani Wanniarachchi, Sita Tarini Clark, Nathan Creber.

**Supervision:** Neil Donnelly, Manohar Bance, Nathan Creber.

**Validation:** Kujani Wanniarachchi, Sita Tarini Clark, Nathan Creber.

**Visualization:** Kujani Wanniarachchi, Sita Tarini Clark, Neil Donnelly, Nathan Creber.

**Writing – original draft:** Kujani Wanniarachchi, Sita Tarini Clark, Neil Donnelly, Manohar Bance, Nathan Creber.

**Writing – review & editing:** Kujani Wanniarachchi, Sita Tarini Clark, Neil Donnelly, Manohar Bance, Nathan Creber.

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
