## [Decision Letter · Decision Letter 0]

1 Sep 2025

Dear Dr. Clark,

Thank you for submitting your manuscript to PLOS ONE. After careful consideration, we feel that it has merit but does not fully meet PLOS ONE’s publication criteria as it currently stands. Therefore, we invite you to submit a revised version of the manuscript that addresses the points raised during the review process.

We look forward to receiving your revised manuscript.

Kind regards,

Toru Miwa

Academic Editor

PLOS ONE

Journal Requirements:

3. Please note that funding information should not appear in the Acknowledgments section or other areas of your manuscript. We will only publish funding information present in the Funding Statement section of the online submission form. Please remove any funding-related text from the manuscript. 

“Professor Bance and the SENSE Lab are supported by the NIHR Cambridge Biomedical Research Centre (NIHR203312*). The views expressed are those of the authors and not necessarily those of the NIHR or the Department of Health and Social Care. Sita Tarini Clark was funded by the Woolf Fisher Trust, New Zealand, the Cambridge Commonwealth, European, & International Trust, and by Trinity College, University of Cambridge. Nathan Creber was funded by the Garnett Passe and Rodney Williams Memorial Foundation.”

5. We note that your Data Availability Statement is currently as follows:

“All relevant data are within the paper and its Supporting Information files.”

Reviewers' comments:

Reviewer's Responses to Questions

**Comments to the Author**

1. Is the manuscript technically sound, and do the data support the conclusions?

Reviewer #1: Yes

Reviewer #2: Yes

Reviewer #3: Yes

2. Has the statistical analysis been performed appropriately and rigorously?

Reviewer #1: N/A

Reviewer #2: Yes

Reviewer #3: Yes

3. Have the authors made all data underlying the findings in their manuscript fully available?

Reviewer #1: No

Reviewer #2: Yes

Reviewer #3: Yes

4. Is the manuscript presented in an intelligible fashion and written in standard English?

Reviewer #1: Yes

Reviewer #2: Yes

Reviewer #3: Yes

Reviewer #1: Authors focus this review in how relevant may perilymph biomarkers be considered for SNHL disorders diagnosis and prognosis, adding results and updates on molecular biology for different SNHL-related disorders published in the last years. This review appears to be well-researched and accurately presented. However, there are some areas where clarity, grammar, and consistency could be improved to enhance readability and comprehension.

The term "sacred" might be too informal or subjective for a scientific paper in this journal. Consider using "critical" or "essential".

In the line 148, "Keywords were divided into two groups" could be expanded to explain the rationale behind the grouping of keywords.

The inclusion of animal studies in the primary search and their subsequent removal at screening might introduce bias or inconsistencies. It would be better to refine the search strategy to accurately exclude animal studies from the beginning or add a paragraph clarifying how animal vs human studies were reliably identified.

In this regard, the use of Google Scholar for identifying target papers before implementing the search might not be comprehensive nor reproducible. It would be beneficial to include other databases or sources to ensure a thorough search.

About the type of studies used for this review, the inclusion of correlational studies might need further clarification. It would be beneficial to specify the type of correlational studies included.

The exclusion criteria list is comprehensive, but it might be helpful to explain the rationale behind excluding certain types of studies, such as "case studies" or "review articles." Same way, in section 2.4, it would be beneficial to explain the rationale behind the selection of specific data points for extraction.

The exclusion of studies based on sample type at the full-text screening stage might need further clarification. It would be helpful to explain why these sample types were excluded and how this impacts the study's findings.

Reviewer #2: The systematic review aims to highlight the gap regarding the utility of perilymph biomarkers

in SNHL diagnosis, treatment, and prognosis within the current literature

With the availability of Perilymph liquid biopsie analyzing tools scientists subsequently have the potential to more accurately characterise inner ear pathologies and provide clinicians with additional diagnostic tools.

Therefore it is essential to" separate the wheat from the chaff" which is to a good extent achieved by this well researched review.

Reviewer #3: though multiple studies were reviewed and summarized in the best possible manner but no comparable biomarkers where there in literature. giving a new classicification for future studies doesn`t make a lot of sense without conducting your own study

**Do you want your identity to be public for this peer review?** For information about this choice, including consent withdrawal, please see our Privacy Policy

Reviewer #1: No

Reviewer #2: No

Reviewer #3: No

---

## [Author Response · Author response to Decision Letter 1]

18 Sep 2025

We would like to thank the reviewers for the encouraging comments and feedback. Please find below our responses to the comments and suggestions made by the reviewers in the documents attached. Best wishes, Sita Tarini Clark

---

## [Editor Report · Decision Letter 1]

26 Sep 2025

Utility of cochlear perilymph biomarkers for hearing loss - a systematic review

PONE-D-25-17529R1

Dear Dr. Clark,

We’re pleased to inform you that your manuscript has been judged scientifically suitable for publication and will be formally accepted for publication once it meets all outstanding technical requirements.

Kind regards,

Toru Miwa

Academic Editor

PLOS ONE
---

## [Editor Report · Acceptance letter]

PONE-D-25-17529R1

PLOS ONE

Dear Dr. Clark,

I'm pleased to inform you that your manuscript has been deemed suitable for publication in PLOS ONE. Congratulations! Your manuscript is now being handed over to our production team.

Kind regards,

on behalf of

Dr. Toru Miwa

Academic Editor

PLOS ONE